# Multi-Channel Generative Framework and Supervised Learning for Anomaly Detection in Surveillance Videos

**DOI:** 10.3390/s21093179

**Published:** 2021-05-03

**Authors:** Tuan-Hung Vu, Jacques Boonaert, Sebastien Ambellouis, Abdelmalik Taleb-Ahmed

**Affiliations:** 1CERI SN, IMT Lille Douai, 941 Rue Charles Bourseul, 59500 Douai, France; jacques.boonaert@imt-lille-douai.fr; 2COSYS Department, LEOST, Gustave Eiffel University, 59666 Villeneuve d’Ascq, France; sebastien.ambellouis@univ-eiffel.fr; 3Opto-Acoustic-Electronics Department, IEMN, CNRS, UMR 8520, Université Polytechnique Hauts de France, 59313 Valenciennes, France; Abdelmalik.Taleb-Ahmed@uphf.fr

**Keywords:** anomaly detection, deep learning, generative model, Conditional GAN, supervised learning, video processing, transportation application

## Abstract

Recently, most state-of-the-art anomaly detection methods are based on apparent motion and appearance reconstruction networks and use error estimation between generated and real information as detection features. These approaches achieve promising results by only using normal samples for training steps. In this paper, our contributions are two-fold. On the one hand, we propose a flexible multi-channel framework to generate multi-type frame-level features. On the other hand, we study how it is possible to improve the detection performance by supervised learning. The multi-channel framework is based on four Conditional GANs (CGANs) taking various type of appearance and motion information as input and producing prediction information as output. These CGANs provide a better feature space to represent the distinction between normal and abnormal events. Then, the difference between those generative and ground-truth information is encoded by Peak Signal-to-Noise Ratio (PSNR). We propose to classify those features in a classical supervised scenario by building a small training set with some abnormal samples of the original test set of the dataset. The binary Support Vector Machine (SVM) is applied for frame-level anomaly detection. Finally, we use Mask R-CNN as detector to perform object-centric anomaly localization. Our solution is largely evaluated on Avenue, Ped1, Ped2, and ShanghaiTech datasets. Our experiment results demonstrate that PSNR features combined with supervised SVM are better than error maps computed by previous methods. We achieve state-of-the-art performance for frame-level AUC on Ped1 and ShanghaiTech. Especially, for the most challenging Shanghaitech dataset, a supervised training model outperforms up to 9% the state-of-the-art an unsupervised strategy.

## 1. Introduction

Abnormal events detection in video sequences is the identification of unpopular actions which produce different features from normal ones. The necessity of automatic anomaly detection in real world context, especially for autonomous driving systems, helps this topic getting more and more attention. Obviously, the differences between anomaly detection and classical action recognition problem are of two types. Firstly, the frequency of abnormal events is really low compared with normal ones, and it leads to an unbalanced scenario where the number of samples in each class is significantly different. Secondly, the features of abnormal events usually do not follow any spatial or temporal relation, so it raises the difficulty of pre-defining the structure or class of abnormal events.

Most of recently successful studies [1,2,3,4,5,6] have tackled this challenge in specific unsupervised ways. They only use normal samples from training sets to generate the standards for normal actions, then try to enlarge the deviation between abnormal samples in the test set and their standards. Using unsupervised strategy naturally follows the structure of popular benchmarks datasets containing only normal samples that do not require annotation task.

The state of the art proposes two approaches for defining the anomalous feature that are, respectively, based on changing detection [1,3,7] and reconstruction/prediction errors [4,5,6]. The first solution is a natural approach where each event is compared with its neighbors to find the most different ones. The weakness of this solution appears when an abnormal event always or never happens in a sequence. Besides, this solution is mainly appropriated for binary classification and it has limitations when we need further information, such as type and localization. The second approach deals with these limitations and achieves state-of-the-art performance. Technically, they generate the future prediction or reconstruct current information for each actions by GANs models and Convolutional Auto Encode-decode (CAE) models. Intuitively, the models trained by only normal samples of the training set will reconstruct better images for normal frames than abnormal ones in the test set. Inspired by the promising results following this approach and the benefits from the rise of CGANs and CAE models, we continue to extend the architecture of reconstructing models by integrating four pix-to-pix Image Translation CGAN models [8] as four parallel channels. This 4-channels framework processes both appearance and motion information in grayscale or color format. Our architecture is presented more specifically in Section 3.1.

In terms of feature extractions and descriptors encoding, previous anomaly detection frameworks proposed various feature types reasoning: at object-centric or local level [2,6,9,10,11], at frame level [4,5], or at both [1]. On the one hand, some methods encode their features to calculate abnormality scores; the decision is done thanks to a threshold or a peak estimation [4,5]. In fact, by integrating all features into one score value, spatial and temporal information are discarded, and a simple comparing decision model is learned. On the other hand, some keep their large scale features [3,6] to train a classifier. If these solutions maintain the wealth of information of the features, the learning process is more complex and takes a long time to converge. To solve this problem, we propose to integrate the complete wealth of the features produced by each of the channels of our framework into a 10-dimensions vectors by PSNR technique. The implementation details will be introduced in Section 3.2. Our descriptors bring specific information of each channel, but are light enough for fast learning models.

In this paper, we first have modified the state-of-the-art unsupervised GAN-based abnormality detection by increasing the feature space. Second, we have added a supervised final step to improve the detection rate. In summary, our contributions are as follows:We introduce a flexible and powerful framework containing a multi-channels CGANs (four streams with nine output channels in our case) to generate multi-type future appearance and motion information. Our architecture considers more consecutive frames forward translation from *t* to (t+2) than previous methods with only encode-decode reconstruction [6] or translation from *t* to (t+1) [4,5]. The number of channels can be freely inserted or removed for multiple purposes.We experimentally prove the effectiveness of PSNR in image comparison task. Based on PSNR method, a useful descriptor of output from our generative framework and ground-truth is proposed. The size of the descriptor is also flexible (equal to the number of channels) and small to adapt to fast classifiers.We demonstrate the improvement of anomaly detection by adding a supervised stage exploiting the feature extractor proposed by our unsupervised architecture. We add a SVM stage that we train on abnormal samples from the test dataset. We achieve at least competitive performances on all benchmarks: Avenue, Ped1, Ped2 in term of frame-level AUC, and a huge improvement on the most challenging datasets, such as Shanghaitech.

The rest of the paper is organized as follows. We summarize related studies on anomaly detection in Section 2. Section 3 describes our architecture in detail. All the experiment implementations and results are presented in Section 4. Finally, we draw final conclusions and future work in Section 5.

## 2. Related Work

In this section, we briefly summary recent successful works in anomaly detection.

### 2.1. Early Works with Hand-Crafted Features

Before Convolutional Neural Networks (CNNs) became popular, most of early methods were based on the extraction of hand-crafted features to estimate the models of normal and abnormal events. Motion trajectories were used as the principal features [12,13] because of their fast extraction and simple implementation. However, the single motion information was not sufficient to represent all the spectrum of abnormal events, and the motion estimator was easily confused in crowed and complex scenes. To improve these limitations, both appearance and motion were extracted along the trajectories. Kim et al. [14] used Histogram of optical flow to build a space–time Markov Random Fields graph. Mahadevan et al. [9] learned the Mixture of Dynamic Textures (MDT) during training then computed negative log-likelihood of the spatio-temporal patch at each region at test phase. Histograms of optical flow orientation (HOFO) were extracted by Wang et al. [15] to classify abnormal events by one-class SVM or kernel PCA. More recently, Giorno et al. [7] built a combination of HOG, HOF, and MBH to train their classifiers, then took the average classification scores to draw the output signal. Generally, most of those methods yielded just moderate performance due to the limitation of hand-crafted features in case of large datasets and complex scenarios.

### 2.2. Recent Successful Models with Deep Learning Features

Recently, the existence of powerful deep learning models leads to many successful approaches in anomaly detection. Hasan et al. [1] learned all motion trajectories features (HOG, HOF, MBH) then built an auto-encoder to reconstruct the scene. The idea of using reconstruction error to measure the regularity score was promising and has been extended by almost all the state-of-the-art methods. Hinami et al. [16] solved the problem of environment-dependent nature by integrating a generic Fast R-CNN model and environment-dependent anomaly detectors. They learned CNN with multiple visual tasks to exploit semantic information that is useful for detecting and recounting abnormal events, and then appropriately plugged the model into anomaly detectors. Ionescu et al. [3] introduced a framework without requirements of training data by applying unmasking techniques. They combined the motion features computed from 3D gradients at each spatio-temporal cube with *conv5* layer of VGG-net with fine-tuning as appearance features. Then, a binary classifier was trained to distinguish between two consecutive video sequences while removing at each step the most discriminant features. The higher training accuracy rates of the intermediately obtained classifiers represented abnormal events. Liu et al. [2] integrated the ConvNet encoding appearance features for each frame and a ConvLSTM memorising motion features for all past frames with auto-encoder to learn the regularity of appearance and motion for the ordinary moments. During this period, they also proposed another work in [11], in which they mapped the temporally-coherent sparse coding, where they enforced similar neighboring frames being encoded with similar reconstruction coefficients with a special type of stacked Recurrent Neural Network (sRNN). At the same time, Liu et al. [4] introduced the first work of future prediction-based anomaly detection. They adopted CGAN techniques with U-Net model as generator to predict the next frame. To generate a high quality image, they made the constraints in terms of appearance (intensity loss and gradient loss) and motion (optical flow loss). Then, the difference between a predicted future frame and its ground truth was used to detect an abnormal event. Developing this approach, Nguyen et al. [5] designed a model as the combination of a reconstruction network and an image translation model that share the same encoder. The former sub-network determined the most significant structures that appear in video frames, and the latter one attempted to associate motion templates to such structures. Then, Ionescu et al. [6] achieved state-of-the-art performance on various popular benchmarks [4,9,10] by building the reconstruction error models that learned both appearance and temporal gradient feature at object-centric levels, then combined K-means clustering with SVMs techniques to produce abnormality scores.

## 3. Proposed Method

We propose a method based on the prediction/reconstruction of the motion and image streams that exploits errors of prediction to detect abnormality. We define a multi-channel framework based on CGAN to produce feature maps for better representing appearance and motion. Unsupervised strategy is used to train the proposed networks and we yield state-of-the-art performance. In a second part of the work, we add a supervised binary SVM as a final layer that takes these feature maps as input after a PSNR transformation. We also present how we extend our model from frame-level detection to pixel level anomaly localization task. Our general pipeline is illustrated in Figure 1.

### 3.1. Multi-Channel pix2pix-CGAN Framework

Our framework combines 4 parallel streams containing 9 channels corresponding to the same pix2pix-CGANs [8] architecture described in Figure 2.

**Pix2pix-CGAN:** Our model contains 8 blocks (i.e., a sequence of layers with the same shape) encode–decode U-Net with skipped connections for the Generator, and 4 blocks for the Discriminator. The Encode block of the Generator and the Discriminator include Convolution, Batch Normalization, and Activation layers. The Decode block is integrated by an extra Dropout layer with a dropout rate of 50%. Let C*k*, CE*k*, and CD*k* denote, respectively, a Discriminator block, an Encoder block, and a Decoder block with *k* filters. The model architecture is defined as:Encoder: CE64-CE128-CE256-CE512-CE512-CE512-CE512-CE512Decoder: CD512-CD512-CD512-CD512-CD512-CD256-CD128-CD64Discriminator: C64-C128-C256-C512

Only the last CE512 block does not have a BatchNorm layer. All convolutional layers of Encoder are applied 4×4 filters with stride 2 for downsampling the input source image to the bottleneck layer. Then, Decoder uses transpose convolutional layers for upsampling from the bottleneck output size to the predicted output size. We also add skip connections between the layers of encode–decode corresponding to the same size of feature maps. The source image is considered as the input of the Generator, and it is concatenated with the target image to produce the first input for Discriminator. The output image of the Generator concatenated with the source image is fed to the Discriminator as second input. We apply L1 loss to measure the distance between target image $I_{t}$ and generated image $I_{g}$ from source image *I*:(1)$$L_{G}=E_{g}{∥I_{t}-I_{g}∥}_{1}$$

The adversarial loss of Discriminator D is calculated by Conditional GAN strategy:(2)$$L_{D}=E_{t}\log D\left(I_{t}|I\right)+E_{g}\log\left(1-D\left(I_{g}|I\right)\right)$$
where $D(I_{t}|I)$ is the discriminator’s estimate of the probability that target image $I_{t}$ is real w.r.t input image *I* (i.e., the image contains $N_{t}+N_{i}$ channels); $D(I_{g}|I)$ is the discriminator’s estimate of the probability that predicted image $I_{g}$ is real w.r.t input *I* (i.e., the image contains $N_{o}+N_{i}$ channels).

The final loss is the sum of both loss with regularization factors $\lambda_{G}$ and $\lambda_{D}$:(3)$$L=\lambda_{D}L_{D}+\lambda_{G}L_{G}$$

**Multi-channel pix2pix-CGAN:** In order to achieve richer and more sensitive features of appearance and motion, we propose 4 parallel pix2pix-CGANs with different input and output configurations, as mentioned in Table 1. We separately investigate the temporal evolution of motion and appearance (in both grayscale and RGB format) by CGAN-1, CGAN-3, and CGAN-4, while CGAN-2 explores the relation between appearance and motion at the same time. Regarding temporal length, CGAN-1 and CGAN-3 learn the short evolution from current frame to its next frame, while CGAN-2 and CGAN-4 study the evolution from *t* to t+2.

Before passing into CGAN streams, all source image channels are resized to 256×256 resolution. We also normalize optical flow maps along both axis to range [0,1] by the following equation:(4)$$F_{m,n}^{norm}=\left\{\begin{array}{cc} \left(F_{m,n}-F_{min}\right)\times \frac{-0.5}{F_{min}}, & \mathrm{for}\;F_{m,n}⩽0\\ F_{m,n}\times \frac{0.5}{F_{max}}+0.5, & \mathrm{for}\;F_{m,n}>0 \end{array}\right.$$
where $F_{m,n}$ is the flow value at pixel (m,n), $F_{max}$, and $F_{min}$ are the maximum and the minimum value of optical flow over all videos. By this way, negative flow values are mapped to [0,0.5], while positive values are mapped to [0.5,1]. Hence, we can maintain the difference between motion directions. All pix2pix-CGAN streams are then trained by Adam optimizer [17] with a learning rate of 0.0002 and momentum parameter $\beta_{1}=0.5$, $\beta_{2}=0.999$ for both Generator and Discriminator. We test with different activation functions (LeakyReLU, ClippedReLU, eLU) and other hyper-parameters such as mini-batch size to find the most suitable parameter values for each stream and dataset.

### 3.2. Feature Extraction with PSNR

In order to measure the distance between two images, we can use two metrics: Mean Square Error (MSE) and Peak Signal to Noise Ratio (PSNR). The work of Mathieu et al. [18] shows that PSNR is a promising way to compare the quality of target (or ground-truth) image $I_{t}$ with generated $I_{g}$:(5)$$PSNR\left(I_{t},I_{g}\right)=10\log_{10}\frac{{\left[max\left(I\right)\right]}^{2}}{\frac{1}{N}\sum_{i=0}^{N}{\left(I_{t}^{i}-I_{g}^{i}\right)}^{2}}$$

Higher PSNR value indicates a better generated image. Obviously, if we accumulate all PSNR values along all channels and all streams to learn a threshold, we might lose the distinction of each channels. So we apply late fusion strategy by separately calculating PSNR for each one. For 4 streams with 9 output channels, we obtain 9 PSNR values to encode as a feature vector. All these vectors are normalized to range [0,1] for all videos sequences. Finally, an abnormal event is detected at frame-level for low values of the global PSNR.

### 3.3. Frame-Level Inference Model

As mentioned in the previous section, we cast anomaly detection problem as a supervised action recognition scenario. We use the unsupervised multi-channel pix2pix CGANs described previously and trained on a dataset containing only regular events. Then we split the test set into two parts: a first part for learning binary SVM classifier and a second part for testing the performance of the trained model. Both parts have abnormal and normal samples. Thus, labeled training data are required. During the evaluation, we analyze how the performance is modified regarding the size of the dataset used to train the SVM classifier. We aim at defining a trade-off between the performance improvement and the amount of work required for the annotation task.

In detail, we have two typical classes for each frame: normal and abnormal frame. Each frame is represented by its feature vector. The full size of feature vector is 9-dimension corresponding to the 9-streams of CGANs. In practical experiments, we also separately test the performance of each CGAN streams and various combinations, so the practical size of feature vector can be from1 to dimension to 9-dimension depending on which combinations are evaluated. Before learning binary SVM classifier, all feature vectors are normalized to [0,1].

### 3.4. Object-Centric Anomaly Localization Model

The objective of this last task is to localize the object related to the abnormal event detected at image-level. For each abnormal frame, we run a fast detector to compute bounding boxes (BB) for each object in the sequence. Each box is considered as a new input and PSNR scores are computed for each one. The BBs yielding minimum values for PSNR scores or values smaller than a threshold refer to the abnormal objects.

To speed up the computational complexity in practical experiments, we do not re-pass each BB throughout whole framework, but we directly apply each BB to error maps. The error map of each frame is the subtract of generated images and its ground-truth after CGAN-streams. The pipeline is illustrated in Figure 3.

Generally, this framework applies a hard-decision strategy to localize the abnormal objects. It leads to the issue that the final performance is highly dependent on the accuracy of bounding box detector. The missing detections or confusing detections can affect the final decision. Another problem is how to pre-define the number of abnormal objects in case of hard-decision. In popular datasets, this number should be one or two object. Then we could search for two BBox candidates obtaining minimum PSNR scores. It based on the fact that the key object in our real world scenario is abnormal one, so the false positive decision is more acceptable than false negative one. We could also go beyond this limitation by searching for a soft threshold of PSNR score.

## 4. Experiments

In this section, we evaluate our unsupervised backbone network with supervised classifier methods on 4 popular datasets for anomaly detection: CUHK Avenue [10], USCD Pedestrian 1-2 [9], and ShanghaiTech [4]. First of all, we briefly introduce those benchmarks and the evaluation metric that we use to measure performance. Then, we present our implementation step by step. Next, we show our promising results in Section 4.3. We end this section by some discussions.

### 4.1. Datasets and Evaluation Metric

**CUHK Avenue:** 16 training sequences with some outliers and 21 testing videos containing 47 irregular events as throwing objects, loitering, and running. The size of the people are changing because of the camera position and angle. The normal samples of the test set are more numerous than abnormal ones.

**USCD Ped1 & Ped2:** Ped1 has 34 training and 36 testing videos with 40 abnormal events. Ped2 is smaller with 16 training and 12 testing videos. Almost abnormal events are related to moving vehicles. Ped1 seems to be more challenging than Ped2 due to the different camera angle. Both have more abnormal events than normal ones on the test set.

**ShanghaiTech:** A very large benchmark containing 13 scenes integrating complex light conditions and camera angles. There are 130 abnormal events and over 270,000 training frames. Moreover, the pixel level ground truth of abnormal events is also annotated. Normal samples are more numerous than abnormal ones on the test set.

**Evaluation metric:** As in the literature, we use a frame-level AUC metric as the main measurement for quantitative evaluation and comparison with state-of-the-art methods. The type of AUC that we applied is AUC of Receiver Operating Characteristic (ROC) curve. A strong classification model has AUC near 1, which means it has a good measure of separability. A weak model has AUC near 0, which means it has the worst measure of separability. In case of AUC=0, the model is predicting all positive samples as negative and vice versa. Particularly, when AUC=0.5, it means the model has not the ability of distinguishing between class or the model is achieving the performance at randomize level. For quantitative evaluation of abnormality localization, we use the same pixel-level AUC and EER metrics reported in [19]. If the intersection between a detected box and the ground-truth box are smaller than 40% of the area of ground-truth box, the detected box are removed.

### 4.2. Implementation Details

**Implementation frameworks:** To build each Pix2pix CGAN stream, we implement our architecture based on Justine Pinkney framework [20] on Matlab. Regarding optical flow extraction, we apply Full Flow [21] on Avenue, Ped1, and Ped2. Because ShanghaiTech is larger than the other datasets, we use the simple Lucas-Kanade [22] optical flow implemented in OpenCV to reduce time processing. After calculating PSNR score for each CGAN stream to encode feature vectors, we learn binary SVM classifier using Classification Learner Toolbox on Matlab. For the object-centric anomaly localization step, we apply the pre-trained model Mask R-CNN [23] as the object detector. All steps are implemented on Matlab R2020a with Nvidia GeForce GTX 1080 environment except Fullflow algorithm that is used on Matlab R2018a.

**Implementation parameters:** For each CGAN stream, we investigate the effect of mini-batch size (M), activation function (A), and number of training epoch (E). We start from E=20 for Avenue, Ped1 and Ped2 with M=32,64,128,256 and A=leakyRELU(α=0.2),clippedReLU(α=0.5),eLU(α=1). Then, we increase E up to 30, 40 and observe the convergence of each loss function to choose the most suitable parameters according to the balance between time consumption and loss convergences. An example is illustrated in Figure 4. Particularly, because ShanghaiTech has a very large training set, we train at only 1 epoch and optimize parameter for CGAN-2, then adapt those parameters for the other CGANs to reduce processing time. We choose CGAN-2 as reference stream for all optimization process because CGAN-2 can learn the relation between both appearance and motion evolution. For inference phase, we split test set into two subsets: SVM-train and SVM-test. To explore the effect of supervised scenario, we start to take from 10% up to 80% from the training set to build SVM-train. We also split SVM-train into train and validation sets for 5-fold cross validation. To optimize the classifier, we run optimization process on SVM learner with different kernel types: Gaussian, linear, cubic, and quadratic. Similarly to the previous phase, on ShanghaiTech dataset, we run the optimization process only once when SVM-train equals 80%. Then we apply optimized parameters for the rest to save time.

**Outliers removing on Avenue:** Several outlier frames exist in Avenue training set. For the frameworks that calculate abnormal scores then compare with threshold [4,5,6], this task is necessary. Because the outlier actions in the training set are annotated as abnormal in the test set. By removing outliers, we avoid the case when a same action is classified as abnormal in the test set, but normal in the training set. Not to do it manually, we train CGAN-2 on the training set to produce output images. Then, we accumulate MSE values between output and ground-truth images for all three channels and we draw the corresponding values. We can easily observe some peaks (Figure 5) occurring at the abnormal frames on training videos. We remove the outlier images corresponding to the peaks to obtain a new clean Avenue training set.

### 4.3. Results

**Quantitative evaluations:** We do the first experiments to evaluate the performance of feature encoding using PSNR and MSE. We choose Avenue benchmark as reference dataset because there are color images with acceptable dataset size (Pedestrian dataset have only grayscale frames and ShanghaiTech is too large). Results are illustrated in Table 2. The large margin about 10% on all streams and combination shows that PSNR technique is significantly better for frame level anomaly detection task.

Next, we apply PSNR technique for the other benchmarks. We go further by investigating various combinations of four streams. Multi-stream is combined by concatenating PSNR features of each stream to produce a new vector. Results are shown in Table 3. On two grayscale datasets, Ped1 and Ped2, CGAN-3 and CGAN-4 produce almost the same performance. The longer temporal evolution of CGAN-4 brings us a small improvement 1∼2% with respect to CGAN-3. On the other RGB datasets, CGAN-3 surpasses CGAN-4 with a huge difference of 10%. Generally, CGAN-2 achieves the best performance among the four streams. It shows that learning both appearance and motion evolution can help us generating better features. Obviously, the combination of CGAN-1 (flow) and CGAN-4 (grayscale) produce similar or slightly better performance than CGAN-2 (combines flow with grayscale). This result also proves that the longer study of temporal evolution at CGAN-4 improves performance. Besides, when we combine several streams, we usually achieve better performance than single stream up to combination of two, three, and all four streams. The fact that the four streams combination always produces the best results is a strong experimental proof of the relevance of our proposed idea about multi-channel framework.

Then, we explore the effectiveness of the supervised layer. By increasing sequentially the SVM-train set size from 10% of the original test set size up to 80%, we report the evolution of performance on Figure 6. Obviously, the larger size of SVM-train set we create, the better performance we obtain.

We compare our best results for each benchmark with recent state-of-the-art methods in Table 4. We split those methods in two groups. The first group contains the fully unsupervised methods without adding extra abnormal samples from original test set to training set. We notice that the method of Ionescu et al. [6] is unsupervised, but they calculated abnormal scores by supervised SVM strategy. The second group contains the semi-supervised and supervised methods that insert abnormal samples into the training set. Generally, it is difficult to comparing the methods in different scenarios, especially in the second group, because the number of testing samples is not the same. Our solution performs almost uniformly and produces promising results on all four datasets. We achieve moderate improvement on Ped1 by about 2% while producing a competitive result on Avenue and Ped2. Particularly, we obtain a large margin of about 9% compared to the state-of-the-art method [6] on the challenging ShanghaiTech dataset. Considering the effect of the supervised scenario, we show that from 50% of the original test set size, we surpass state-of-the-art performance on Ped1 and ShanghaiTech.

For the abnormal object localization models, we evaluate our quantitative performances in Avenue dataset using AUC and ERR metrics at pixel-level in Table 5. Most of state-of-the-art researches [4,5,6] have not reported quantitative performances for abnormality localization task but only qualitative analysis. Hence, we compare our quantitative results with recent methods of Vu et al. [19,26] that applied the same evaluation metrics. We achieve significant improvements on both metrics.

**Qualitative evaluation:** To understand the distinction of our features on each stream, we explore the distribution of test samples for each CGAN on each dataset. Figure 7 shows us the distribution of test samples in Ped2 dataset. Obviously, almost normal samples situate in higher values than abnormal ones. This distribution corresponds to the fact that PSNR techniques produce high scores if predicted images tend to similar source images. Besides, because Ped2 contains only grayscale images, all three channels of CGAN-3 must be equal so we observe a straight line distribution.

Figure 8 present several examples selected from Avenue, Ped1, Ped2, and ShanghaiTech to demonstrate true positive and failure cases of object-centric anomaly localization. Obviously, we can strongly detect various types of abnormal events even in crowed and occlusive background with multiple camera angle, involving not only single, but also multiple objects. Many failure cases appear when abnormal objects are too close to the other normal boxes then PSNR score of those boxes can be effected by the features of abnormal objects. Other common errors are directly caused by false negative frame-level detection at previous phase.

### 4.4. Discussion

**Strengths:** Our experiments show that we achieve very promising results, although we have not optimized all of the steps. Suppose that we replace Full-Flow [21] and Lucas-Kanade [22] by recent state-of-the-art methods, we might improve our performance. Besides, our framework is quite flexible, so we can add more streams with more channels representing various types of data to learn richer features. Those features are also appropriated for many classifiers.

**Limitations:** On one hand, the mix of unsupervised CGAN streams with supervised SVM help us obtaining impressive performance on the offline case, where all test data are prepared and well pre-processed. For online works, we get trouble when brand new abnormal actions suddenly occur without existing in the previous learned database. On the other hand, the anomaly localization is significantly affected by the detector we use (Figure 9). Sometimes, bounding boxes is imperfectly generated with size and labels. We can go beyond this limitation by not applying a high-level detector such as Mask R-CNN, but only a low-level region proposal network.

## 5. Conclusions and Future Work

In this paper, we present a flexible multi-channel generative framework for supervised anomaly detection in surveillance videos. Various types of input images are passed into 4 CGAN streams to predict future information. Then we apply PSNR technique to encode prediction error into feature vectors. A supervised scenario is built based on a part of the test set to train a SVM classifier. Our solution is evaluated on 4 datasets: Avenue, Ped1, Ped2, ShanghaiTech, and surpass state-of-the-art performance on three of them, especially producing a huge improvement on ShanghaiTech.

For future work, we first plan to replace optical flow estimation methods as discussed above. The USCD Ped2 is relatively easy, so we will evaluate our methods with more datasets [27,28,29]. Some of those datasets are suitable for supervised scenario. Then, we would like to build a new dataset for supervised anomaly detection containing both normal and abnormal samples in its training set. Furthermore, we would like to build an end-to-end model without adding SVM classifiers.

## Figures and Tables

**Figure 1 sensors-21-03179-f001:**
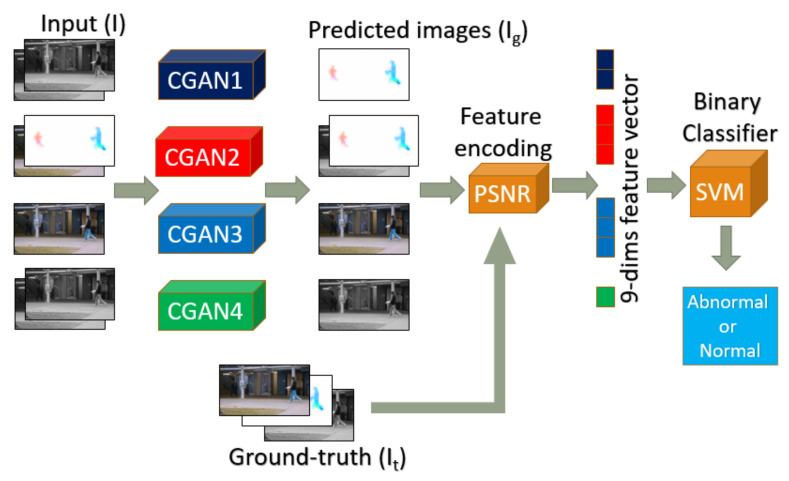
Our multi-channels pix2pix-CGANs framework for anomaly detection. In each CGAN stream, the number of channels and the type of each channel for input image (i.e., source image *I*) and output image (i.e., generated or predicted image $I_{g}$) are different. CGAN1 takes 2 grayscale channels as input and 2 optical flow channels as output. CGAN2 takes 1 grayscale channel and 2 optical flow channels as input, then 1 grayscale channel and 2 optical flow channels as output. CGAN3 takes 3 RGB channels as input and 3 RGB channels as output. CGAN4 takes 2 grayscale channels as input and 1 grayscale channel as output. The configuration is described in detail in Table 1. The channels of generated images are similar to the channels of ground-truth images (i.e., target images $I_{t}$). Best viewed in color.

**Figure 2 sensors-21-03179-f002:**
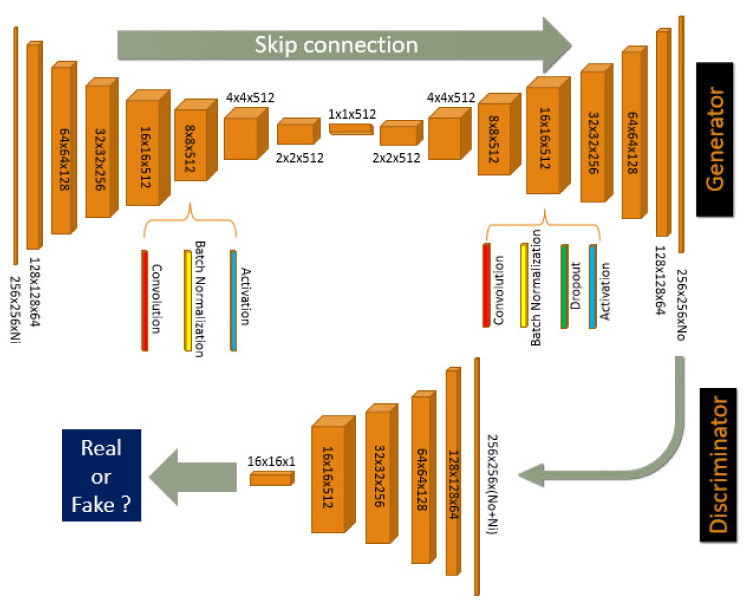
Our pix2pix-CGAN architecture. Encoder blocks CEk are from first block $256\times 256\times N_{i}$ to bottleneck block 1×1×512 of Generator. Decoder blocks CDk are from bottleneck block 1×1×512 to last block $256\times 256\times N_{o}$ of Generator. $N_{i}$, $N_{o}$ denote the number of input and output image channels of Generator. Discriminator blocks Ck are from block $256\times 256\times (N_{i}+N_{o})$ to block 16×16×1. Because of the skip connection, the total channels that pass to Discriminator is $N_{i}+N_{o}$. Best viewed in color.

**Figure 3 sensors-21-03179-f003:**
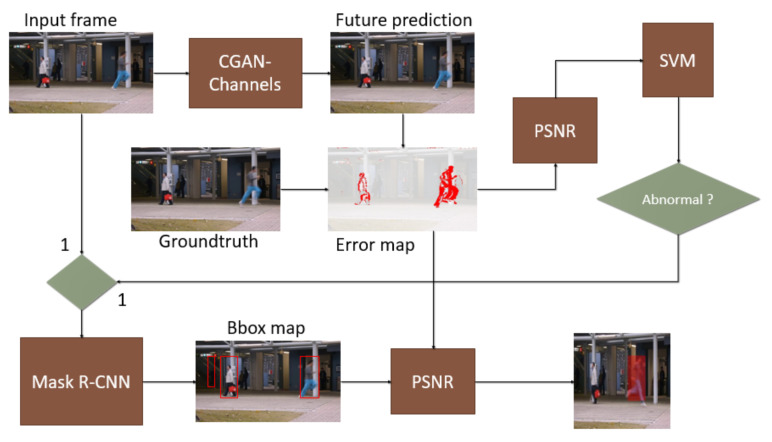
The pipeline of object-centric anomaly localization framework. After the frame-level anomaly detection stage, we obtain the labels for each frame. Only positive samples (i.e., abnormal frame) are taken into account to localization stage. For each abnormal frame, we run a fast detector to compute bounding boxes (BB) for each object in the sequence. To speed up the computational complexity in practical experiments, we do not re-pass each BB throughout whole framework but we apply directly each BB to error maps. PSNR scores are computed for each one. The BBs yielding minimum values for PSNR scores or values smaller than a threshold refer to the abnormal objects.

**Figure 4 sensors-21-03179-f004:**
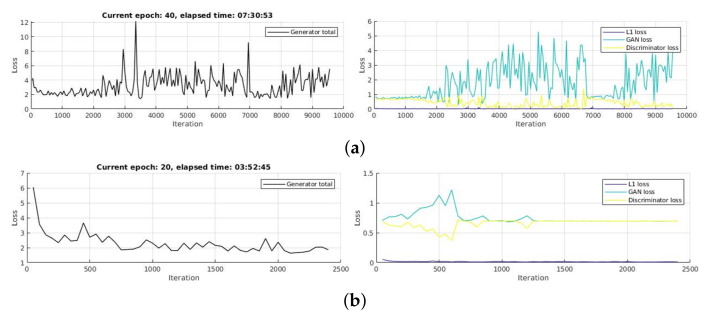
Comparison of loss convergence between different paramenters for CGAN-2 on Avenue dataset. (**a**) shows the case where the losses are not convergent at 40 epochs, Minibatch size 64 and leakyReLU activation function, while (**b**) illustrates a good set of parameters, E=20,M=128,A=eLU, corresponding to good loss convergence. We find that the total loss is stable from 1000th iteration, and each loss is almost convergence. Best viewed in color.

**Figure 5 sensors-21-03179-f005:**
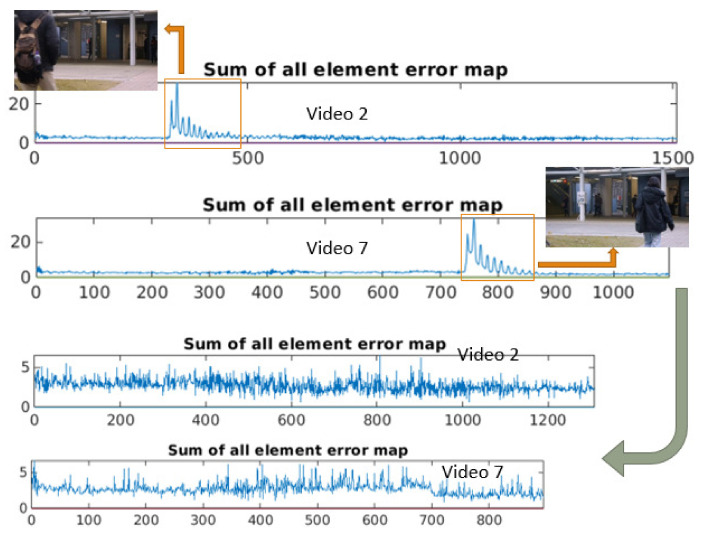
Example of outliers removing on Avenue dataset. Not to do it manually, we train CGAN-2 on training set to produce output images. Then we accumulate MSE values between output and ground-truth images for all 3 channels and we draw the corresponding values. We can easily observe some peaks occurring at the abnormal frames on training videos. We remove the outlier images corresponding to the peaks to obtain a new clean Avenue training set. Best viewed in color.

**Figure 6 sensors-21-03179-f006:**
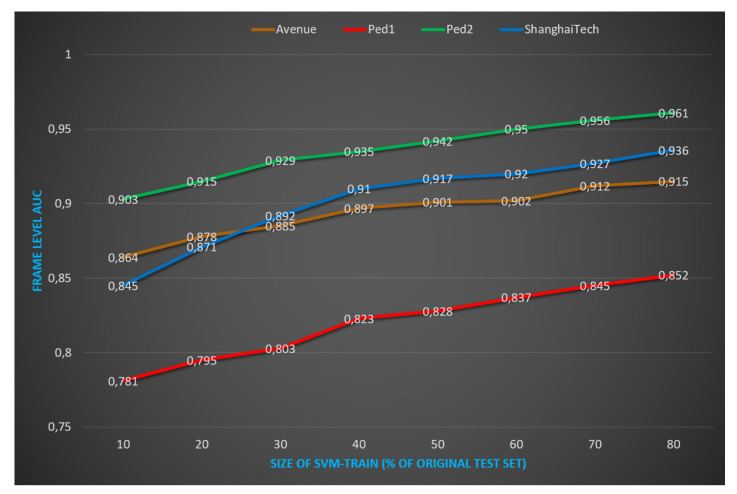
Evolution of AUC performance according to the size of SVM-train set. The vertical axis shows the AUC performances and the horizontal axis shows the size of SVM-train set corresponding to how many samples of the original test set. By increasing sequentially the SVM-train set size from 10% of the original test set size up to 80%, we report the evolution of performance. Obviously, the larger the size of SVM-train set we create, the better performance we obtain. Considering the effect of supervised scenario, we show that from 50% of the original test set size, we surpass state-of-the-art performance on Avenue, Ped1, and ShanghaiTech. Best viewed in color.

**Figure 7 sensors-21-03179-f007:**
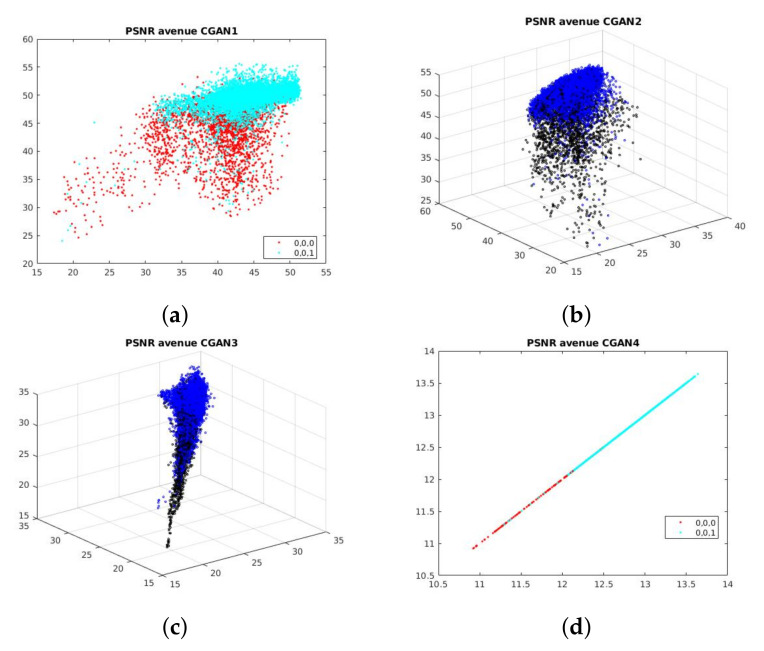
Distribution of normal and abnormal samples in Avenue dataset. For CGAN-2 (**b**) and CGAN-3 (**c**), normal points are blue and abnormal points are black. For CGAN-1 (**a**) and CGAN4 (**d**), normal = light blue and abnormal = red. Obviously, normal samples situate in higher values than abnormal ones. This distribution corresponds to the fact that PSNR techniques produce high scores if predicted images tend to similar source images. On one side, because Avenue contains RGB images, CGAN-3 has 3-dimensions values.

**Figure 8 sensors-21-03179-f008:**
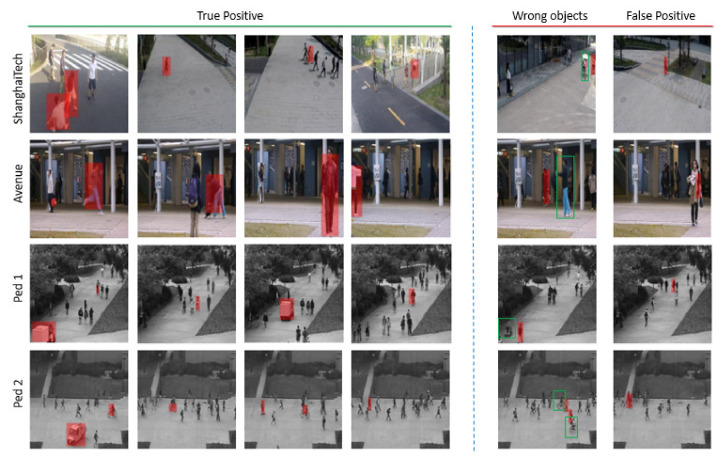
Qualitative results of abnormal objects localization on Avenue, Ped1, Ped2, and ShanghaiTech dataset. The left side contains correct localizations and the right side shows the failure cases with two main errors: false negative frames and detect wrong objects. The true objects are marked by green boxes. Best viewed in color.

**Figure 9 sensors-21-03179-f009:**
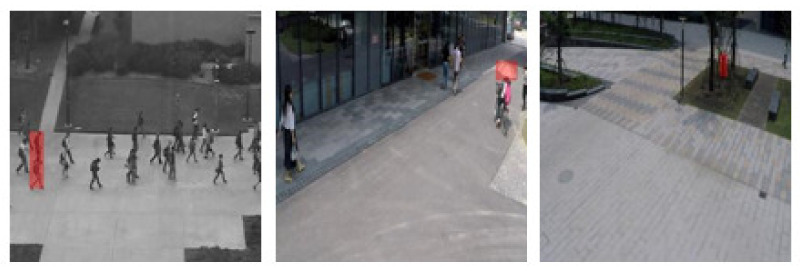
Several imperfect cases of anomaly localization affected by Mask R-CNN detector. **Left**: Too large bounding box; **Middle**: Too small box; **Right**: Wrong object. Best viewed in color.

**Table 1 sensors-21-03179-t001:** Configuration of input *I*, output (i.e., predicted images $I_{g}$) and target images $I_{t}$ for each pix2pix-CGAN stream. $N_{i}$, $N_{o}$, $N_{t}$ denote the number of input, output, and target image channels. In all the cases, the channels of output and target images must be similar. $G_{t}$, $F_{t\to t+1}^{x,y}$, and $RGB_{t}$ equal to grayscale image at frame *t*, optical flow from frame *t* to t+1 along axis *x*, *y*, and RGB color frame at frame *t*. All channels are taken into account for calculating loss functions.

Channel	Input (I)	$N_{i}$	Output $I_{g}$ & Target $I_{t}$	$N_{o}=N_{t}$
CGAN-1	$G_{t}$, $G_{t+1}$	2	$F_{t\to t+1}^{x,y}$	2
CGAN-2	$G_{t}$, $F_{t\to t+1}^{x,y}$	3	$G_{t+1}$, $F_{t+1\to t+2}^{x,y}$	3
CGAN-3	$RGB_{t}$	3	$RGB_{t+1}$	3
CGAN-4	$G_{t}$, $G_{t+1}$	2	$G_{t+2}$	1

**Table 2 sensors-21-03179-t002:** Frame-level AUC performance comparison between two methods of prediction error encoding: PSNR and MSE. Results are reported on Avenue dataset using SVM classifiers. 80% samples of test set are used for training SVM.

CGAN Stream	MSE	PSNR
CGAN-1	0.72	0.82
CGAN-2	0.69	0.81
CGAN-3	0.73	0.79
CGAN-4	0.62	0.67
CGAN-(1 + 2 + 3 + 4)	0.81	0.92

**Table 3 sensors-21-03179-t003:** Frame-level AUC performance on all 4 benchmarks using PSNR encoding and SVM classifiers. 80% samples of the test set are used for training SVM. SHT = ShanghaiTech.

CGAN Stream	Avenue	Ped1	Ped2	SHT
CGAN-1	0.82	0.75	0.89	0.68
CGAN-2	0.81	0.78	0.92	0.78
CGAN-3	0.79	0.64	0.75	0.73
CGAN-4	0.67	0.65	0.77	0.62
CGAN-(1 + 2)	0.83	0.80	0.93	0.80
CGAN-(3 + 4)	0.86	0.70	0.77	0.81
CGAN-(1 + 4)	0.86	0.79	0.93	0.76
CGAN-(1 + 2 + 3)	0.86	0.83	0.95	0.87
CGAN-(1 + 2 + 4)	0.83	0.83	0.93	0.83
CGAN-(1 + 2 + 3 + 4)	0.92	0.85	0.96	0.94

**Table 4 sensors-21-03179-t004:** Comparison of Frame level AUC on 4 datasets between ours solution and recent state-of-the-art methods that reported their performance on at least 2 similar datasets. The best performance on each dataset is highlighted by bold number. We split those methods in two groups. The upper group contains the fully unsupervised methods without adding extra abnormal samples from original test set to training set. The second group contains the semi-supervised and supervised methods that insert abnormal samples into training set. Generally, it is difficult to comparing the methods in different scenarios, especially in second group, because the number of testing samples is not the same. For our case, all the output channels of our model are used. 80% samples of test set are used for training SVM. SHT = ShanghaiTech.

	Methods	Avenue	Ped1	Ped2	SHT
Unsupervised methods
	Luo et al. [2]	0.77		0.88	
	Nguyen et al. [5]	0.87		0.96	
	Hinami et al. [16]	0.89		0.92	
	Vu et al. [19]	0.72	0.82	**0.99**	
	Ouyang et al. [24]	0.89		0.97	0.81
	Ionescu et al. [6]	0.90		0.98	0.85
Semi and supervised methods
	IVC with OS [25]	0.83			0.56
	IVC with OS & FL [25]	0.83			0.50
	IVC with OS & FL &2streams [25]	0.81			0.50
	TripleLoss + OCSVM [25]	0.80			0.50
	Hasan et al. [1]	0.80	0.75	0.85	0.61
	Luo et al. [11]	0.82		0.92	0.68
	Ionescu et al. [3]	0.81	0.68	0.82	
	Liu et al. [4]	0.85	0.83	0.95	0.73
	Liu et al. [25]	**0.93**			0.77
	Ours	0.92	**0.85**	0.96	**0.94**

**Table 5 sensors-21-03179-t005:** Pixel-level AUC and EER performance of abnormal object localization on Avenue dataset. The best performance on each dataset is highlighted by bold number.

Methods	AUC	EER
OC-SVM [26]	33.16	47.55
GMM [26]	43.06	43.13
Multilevel Representations [19]	52.82	38.83
Ours	**74.43**	**30.21**

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
