# Peer review of "Multi-Channel Generative Framework and Supervised Learning for Anomaly Detection in Surveillance Videos"

_sensors, 2021, doi:10.3390/s21093179_

Round 1

Reviewer 1 Report

Positives

  1. The problem addressed in this work is very relevant and authors use state-of-the-art GAN network to find the anomaly in surveillance video.
  2. Automatic video surveillance can be highly impactful if achieved with high performance.
  3. Multi-channel CGANs seem powerful to encode critical information.
  4. Results are  really superior when compared to previous related works.

Negatives

     1. Authors didn't try end-to-end NN that could have performed better instead of using SVM in the outer layer.

Author Response

Dear all,

We would like to thank the reviewer for their valuable comments and suggestions. We have taken into account all the suggestions of the reviewer and improved our manuscript corresponding to their useful advices. Please find in attached file our detailed responses for each of your comments.

Best regards,

Reviewer 2 Report

This paper presents a method to detect abnormal events in videos using a multi-GAN approach in conjunction with supervised learning (SVM). There are some important aspects to be addressed, as follows:

1- The idea of combining unsupervised learning with some level of supervised learning is interesting. However, I am concerned that the comparisons made against other SOTA methods are not fair. In essence, other SOTA methods do not use labels to learn to differentiate between normal and abnormal frames. They purely rely on training data with no abnormal events. By using an SVM trained with such abnormal labels, results cannot be directly compared. The authors should emphasize this and additionally, they should provide results when no SVM is used. In other words, results showing how their method works when only using the reconstruction error of the different GANs. Such results are comparable to the ones attained by the SOTA.

2- In Eq. (2), it is not clear what I, I_t, and I_g are in relation to the inputs/outputs tabulated in table 1. For example, for CGAN-1, the inputs are two gray level images ant the output is the optical frame (in x, y-direction). In t this case, what do I, I_t, and I_g represent for the corresponding loss as specified in Eq. (2)? Please improve Table 1  by explicitly specifying which of the inputs/outputs used by each GAN represent I, I_t, and I_g.

3- There are no quantitative elevation results for the object-level anomaly detections. Please provide these.

4- Please clearly specify if the results tabulated in Tables 2 and 3  are computed using the SVM. if so, how much training data (% out of the test data)  was used to train the SVM for these results?

5-  Please clearly specify for the results tabulated in Table 4 if the output of all GANS were used.

6-  The USCD dataset is relatively easy. The authors should include more challenging datasets with more realistic events. Please include results for the following datasets:

* UCF-Crime dataset: Sultani, W., Chen, C., & Shah, M. (2018). Real-world anomaly detection in surveillance videos. In Proceedings of the IEEE Conference on Computer
Vision and Pattern Recognition (pp. 6479{6488).

* Crowd Violence dataset: Hassner, T., Itcher, Y., & Kliper-Gross, O. (2012). Violent flows: Real-time detection of violent crowd behavior. In 2012 IEEE Computer Society Conference on Computer Vision and Pattern Recognition Workshops (pp. 1{6).

* LV Dataset: Leyva, R., Sanchez, V., & Li, C.-T. (2017b). The lv dataset: A realistic surveillance video dataset for abnormal event detection. In 2017 5th International Workshop on Biometrics and Forensics (IWBF) (pp. 1{6). IEEE.

7- Table 4 is missing some SOTA methods evaluated in these datasets. please include these methods:

* Hung Vu et al. “Robust Anomaly Detection in Videos Using Multilevel Representations”. In: Association for the Advancement of Artificial Intelligence (AAAI). 2019

* Reference [16] --> R. Hinami et al. “Joint Detection and Recounting of Abnormal Events by Learning Deep Generic Knowledge”. In: IEEE International Conference on Computer Vision (ICCV). 2017, pp. 3639– 3647

* Y. Ouyang et al., "Video Anomaly Detection by Estimating Likelihood of Representations," sIn: International Conference on Pattern Recognition (ICPR),  2021

*Wen Liu et al. “Margin Learning Embedded Prediction for Video Anomaly Detection with A Few Anomalies”. In: Proceedings of the Twenty-Eighth International Joint Conference on Artificial Intelligence (IJCAI). International Joint Conferences on Artificial Intelligence Organization, 2019, pp. 3023–3030 

8- Have the other evaluated methods also used the Avenue dataset after outlier removal? Please indicate. If not, results may not be directly comparable as dealing with such outliers is more challenging than manually removing them.

9- "On the other side, CGAN-1, CGAN-2, and CGAN-3 streams samples are clearly separated into two clusters while CGAN-4 samples are
more disordered." --> This is not evident from Fig. 7 at all. Visually, it is easy to see the two clusters because the authors have used different colors to depict them, but in terms of having clearly separated clusters, this is not the case. For example, if it were not for the colors used, it is impossible to see the two clusters for the case of "PSNR avenue CGAN1." Please either remove this figure or find alternative ways to discuss these ideas.

10- Finally, please proofread the text to correct typos and grammatical inconsistencies. Some examples are:

*Base on PSNR...
*...of channels and quite small...
*We summary related...
*...integrated a ConvNet encoding appearance features... (more than one feature? then no "a' is needed, check for other similar cases)
*The Decode block is integrated an extra...
*Only the last CE512 block has not BatchNorm layer...
*Figure (a) shows the case where the loss are not convergent at 40 epochs,...
*...iteration and each loss are almost convergence....

Author Response

(The authors gave the same response as above.)

Round 2

Reviewer 2 Report

Thanks for addressing the comments in such a short time. The paper is now ready.